# A Surface-Enhanced Raman Scattering Substrate with Tunable Localized Surface Plasmon Resonance Absorption Based on AgNPs

**DOI:** 10.3390/s24175778

**Published:** 2024-09-05

**Authors:** Guanzhou Lin, Meizhang Wu, Rui Tang, Bo Wu, Yang Wang, Jia Zhu, Jinwen Zhang, Wengang Wu

**Affiliations:** 1National Key Laboratory of Advanced Micro and Nano Manufacture Technology, Beijing 100871, China; 2School of Integrated Circuits, Peking University, Beijing 100871, China; 3School of Instrument Science and Opto-Electronics Engineering, Beijing Information Science and Technology University, Beijing 100096, China; 4School of Automation, University of Science and Technology Beijing, Beijing 100083, China; 5School of Electronic Engineering and Computer Science, Peking University, Beijing 100871, China; 6Frontiers Science Center for Nano-Optoelectronics, Peking University, Beijing 100871, China

**Keywords:** Ag nanoparticles, LSPR, SERS

## Abstract

In this paper, a three-layer structure of silver particle (AgNP)-dielectric-metal is proposed and constructed based on the characteristics of AgNPs that can excite LSPR (Localized Surface Plasmon Resonance) in free space. In order to overcome the problem of AgNPs easily oxidizing in the air, this paper synthesizes AgNPs using the improved Tollens method and effectively suppresses the coffee-ring effect by changing the solution evaporation conditions, so that the distribution of AgNPs in the deposition area is relatively uniform. The structure proposed in this paper takes advantage of the flexibility of nanoparticle application. The AgNPs deposited on the dielectric layer can effectively localize energy and regulate the LSPR of the device well. The structure can not only achieve precise regulation of the LSPR resonance peak of AgNPs but also can be used as a SERS substrate.

## 1. Introduction

The applicability of surface plasmons (SPs) has been extensively demonstrated in the emerging field of nanoplasmonic photonics [1]. Among them, Localized Surface Plasmon Resonance (LSPR) is a phenomenon based on metal nanoparticles excited under specific illumination conditions. It is affected by the size and composition of the nanoparticles and the surrounding environment and has significant influence on electrical constants. LSPR is significantly influenced by the size and composition of nanoparticles, as well as the dielectric constant of the surrounding environment. LSPR also possesses excellent wavelength tuning capabilities. By precisely designing metal structures at the micro- and nanoscale, it is possible to control LSPR characteristics such as resonance frequency, intensity, and spatial distribution effectively, enabling customized design of optical properties and further expanding its range of applications [2,3,4,5].

SERS (Surface-Enhanced Raman Scattering) offers high sensitivity and mild detection conditions, even enabling single-molecule detection [6,7,8]. The applications of SERS-related substrates have made significant progress in various fields, making it crucial to study the interactions between molecules and SERS-active substrates. Today, SERS has become a powerful analytical tool in chemical and biochemical detection, environmental monitoring, and food safety applications [9]. LSPR can produce local field enhancement at the nanoscale, making LSPR a primary mechanism for designing SERS substrates [10]. The surface of metal nanoparticles can interact with incident light to excite LSPR, thereby enhancing the local electric field significantly at the surface of the metal nanoparticles. As a result, more research on SERS focuses on developing substrates with strong electromagnetic field enhancement effects, leading to extensive research and application of LSPR-based SERS substrates [11,12,13].

As a common metallic material, silver has excellent surface plasmon properties in the visible light band. The LSPR response of AgNPs depends on their size, uniformity, shape, and dispersion and depends on the dielectric constant of the surrounding medium [14,15,16]. Therefore, AgNPs are often used as the particle-structure-constituting sensors. The LSPR effect based on AgNPs has been widely researched and applied in many fields including optical antennas, sensors, optical modulators, and photodetectors [17]. By controlling the size and shape of the particles, the properties of the LSPR can be tuned to meet the requirements of a specific device. In terms of SERS detection, AgNPs also show strong application potential. Although gold has better chemical stability and greater biocompatibility than silver, silver has more enhancement and is more effective [18].

However, due to the relatively active chemical properties of silver, individual AgNPs will be rapidly oxidized when exposed to air, which will seriously affect their surface plasmon properties and further affect their related physical effects [17]. Therefore, it is very important to develop AgNPs with good performance, uniform morphology, and strong stability for studying LSPR properties. In practical applications, AgNPs can self-assemble into SERS substrates on different substrates [19]. Among them, the successful deposition on glass has expanded the application of AgNPs, overcome its lack of flexibility in regulation, and achieved precise control of device response [20,21]. However, for the LSPR effect of AgNPs, although different shapes can have different LSPR characteristics, they still lack flexibility in regulation and cannot accurately control the response of AgNPs. Therefore, it is of great significance to expand the application of AgNPs by referring to the device structure.

In this paper, a three-layer structure of AgNP-dielectric-metal is proposed (as shown in Figure 1). AgNPs were synthesized using the improved Tollens method. The AgNPs synthesized by this method are wrapped with a layer of glucose on the outside, which can not only ensure the performance of AgNPs but also effectively prevent the oxidation of glucose and have good stability. Then, gold and silicon dioxide (SiO_2_) are deposited on a glass substrate. The bottom layer of gold acts as a metal reflective layer, and the middle SiO_2_ acts as a dielectric layer. On the top of SiO_2_, a perforated tape is used to limit the deposition range, and AgNP colloids are synthesized by dripping and drying successively, which suppresses the coffee-ring effect of AgNPs, so that the particles can be deposited to form functional areas uniformly. When the top layer of nanoparticles is added, AgNPs can be effectively assembled into a SERS substrate; at the same time, the top layer of AgNPs, the dielectric layer, and metal layer form a three-layer structure. The LSPR response of AgNPs is coupled with the cavity mode of the structure, localizing the electric field energy in the dielectric layer, generating a resonant absorption peak. When the thickness of the SiO_2_ layer is adjusted, the position of the resonance absorption peak can be adjusted precisely within a certain range. The method proposed in this study effectively overcomes the problem of the impact of air oxidation on silver nanoparticles (AgNPs) and avoids the complicated processing process.

## 2. Materials and Methods

### 2.1. Materials

In this section, the reagents used are as follows: silver nitrate (AgNO_3_ 99.9%), D-glucose, ammonia (NH_3_•H_2_O, 98%), Rhodamine 6G (C_28_H_31_N_2_O_3_Cl, R6G), acetone, and ethanol, supplied by Beijing Chemical Works. All the reagents used in this work were analytical-grade reagents. Deionized water was produced by Peking University and used in all experiments.

### 2.2. Fabrication

The metal evaporation in this article is completed by the DE 400 electron beam evaporation coating system. In this article, the evaporated metal is gold, the rate is 2 Å/s, and the deposition thickness is 50 nm. In this chapter, the sputtering of silicon dioxide uses the MSP-3200 fully automatic magnetron sputtering coating machine system, whose RF power is 1000 W, and the deposition rate of SiO_2_ is 0.4 Å/s.

A scanning electron microscope (SEM) was used to characterize AgNPs. The SEM used was FEI’s Nova-NanoSEM 430, whose electron gun acceleration voltage is 200 V–30 kV, whose electron beam current can reach 100 nA, and whose resolution can reach up to 1 nm. The acceleration voltage used was 10 kV–15 kV, which was used for the morphological characterization of the samples. The spectrometer used in this chapter is the ARM microscopic angle-resolved spectrometer of Ideaoptics (Shanghai, China), whose light source is a halogen light source with a power of 100 W and a resolution of 1 nm.

In this study, the Raman spectrometer used for SERS spectra is Horiba JY LabRAM HR Evolution (Paris, France). This Raman spectrometer has a high resolution at about 1 cm^−1^ and has two gratings of 600 nm and 1800 nm, with a spectral range of 200 nm to 2100 nm. The Raman spectrometer uses a 600 nm grating, a laser power of 3.7 mW, an excitation wavelength of 633 nm, and a 10× objective lens to measure the Raman spectrum. All Raman spectra were tested at room temperature.

### 2.3. Synthesis of AgNPs

The Ag nanoparticles are synthesized using a modified Tollens method, which uses the aldehyde group of glucose molecules to reduce silver ammonia complexes to form silver nanoparticles. The principle of the method is as follows [20]:Ag[NH_3_]_2_^+^[aq] + RCHO[aq] → Ag[s] + RCOOH[aq]

The specific synthesis method is as follows: (1) Prepare silver ammonia complex: prepare 0.05 M AgNO_3_ solution with deionized water and stir thoroughly; (2) Add ammonia dropwise with stirring, and the solution will first become turbid and then clear. When the solution becomes clear, stop adding ammonia, and a silver ammonia complex is formed; (3) Prepare 0.4 M glucose solution; (4) Add 50 μL of the silver ammonia complex synthesized in step 2 to the 0.4 M glucose solution; (5) Add the silver ammonia complex once every 30 min, and repeat 4 to 6 times. The solution turns yellow, and glucose-coated AgNPs are obtained. The particles synthesized using this method not only demonstrate the LSPR properties of AgNPs effectively but also provide protection against rapid oxidation from air exposure, which can ensure the LSPR properties of AgNPs in this research [22]. See Figure 2.

### 2.4. Fabrication of the Device

The preparation process of AgNP-dielectric-metal structure is as follows: (1) Clean the substrate and prepare a standard glass substrate; then, clean it in acetone, alcohol, and deionized water for 2 min in an ultrasonic environment to remove any possible organic contaminants; (2) Use electron beam evaporation to sputter 50 nm of gold to form a reflective layer. Before sputtering the gold, 5 nm of titanium needs to be sputtered as an adhesion layer; (3) Sputter 190 nm of silicon dioxide above the gold; the silicon dioxide serves as a dielectric layer; (4) After constructing the dielectric layer, punch holes in the insulating tape and stick it on the dielectric layer; (5) Deposit the synthesized AgNPs in the holes of the tape, adding 40 μL each time, and then, place it on a hot plate at 85 °C to bake to remove moisture; (6) Repeat several times until flat particle gel is formed in the holes; this achieves the preparation of the functional area; (7) Remove the tape to complete the preparation of the AgNP-dielectric-metal structure.

In this study, when AgNPs synthesized from glucose are deposited on the dielectric layer, it is necessary to suppress the coffee-ring effect generated during the colloid deposition process so that the AgNPs are evenly distributed in the functional area, thereby effectively forming relatively uniform discrete AgNPs [23]. The perforated tape has two main functions. One is to limit the spatial distribution of the AgNP colloid and only allow it to be distributed in a specific area; the second function is to suppress the coffee-ring effect of AgNPs [24,25]. During the deposition process, as the water evaporates, the surface tension of the liquid shrinks the liquid to the middle, and the solute will not be evenly distributed in the droplet area [26,27]. Therefore, this study used a hot plate to evaporate the AgNP colloid at a temperature of 85 °C, while continuously adding AgNP colloid to the holes at a dose of 40 μL. In this process, the hot plate can speed up the evaporation rate of the liquid in the AgNP colloid and reduce the difference in evaporation rates between the center and edge areas; during the evaporation process, AgNPs are continuously added dropwise to reduce the flow of liquid caused by evaporation in the edge areas. Since AgNPs are formed by the reduction in glucose, a reduction in water content leads to an increase in the colloid’s viscosity, which in turn restricts the movement of particles during the liquid’s evaporation process. This method can effectively avoid the coffee-ring effect and make the AgNPs have a relatively uniform distribution in the functional area.

## 3. Results and Discussion

### 3.1. The Influence of Dielectric Layer Thickness

In this study, the LSPR is excited by the AgNPs on the top layer. The AgNPs on the top layer, the gold film on the bottom layer, and the dielectric layer in the middle together form a structure similar to the F-P cavity. The distance between the two is determined by the thickness of the dielectric layer in the middle. Figure 3a shows the influence of the thickness of SiO_2_ on the structural response calculated by simulation. Among them, the thickness of the gold film on the bottom layer is 50 nm, and the thickness of the dielectric layer SiO_2_ in the middle layer varies from 50 nm to 350 nm; the diameter of the AgNPs on the top layer is 40 nm, and the spacing between them is set to 2 nm, which simulates the state of close packing of nanoparticles. When the thickness of SiO_2_ is less than 150 nm, the parts of the structure cannot be well coupled. When the thickness of SiO_2_ exceeds 150 nm, the F-P cavity mode of the structure and the LSPR mode of the top layer can be effectively coupled. Starting from 150 nm, the resonance peak steadily redshifts as the thickness of SiO_2_ increases. As the thickness of SiO_2_ continues to increase, the resonance peak approaches the position it occupies at about 150 nm, once it reaches 350 nm. This is because when the thickness of SiO_2_ increases, higher-order modes appear. Figure 3b,c show the simulation results of the electromagnetic field distribution of two mutually perpendicular incident lights at the absorption peak of the structure. It can be seen that under the excitation of the incident light, the LSPR of AgNPs is effectively excited. At the same time, due to the presence of the AgNPs, the electric field energy is effectively localized in the dielectric layer, forming a response similar to the cavity mode.

Through simulation, it can be seen that the height of SiO_2_ has a significant effect on the absorption peak of the AgNP-dielectric-metal structure. The LSPR of AgNPs and the cavity mode response of the structure are coupled to form a resonant absorption peak. When the thickness of the dielectric layer is adjusted, the resonant absorption peak regulates the resonant response of the structure in a fairly large range, and the position of the resonant peak can also be precisely adjusted.

### 3.2. Effect of Nanoparticles

For spherical AgNPs, different nanoparticle sizes will also affect the structural response. When the diameter of the AgNPs varies from 10 nm to 50 nm, the position of the resonance peak redshifts with the increase in the diameter of the AgNPs, but the absorption intensity remains basically unchanged (Figure 4a).

In addition, the distribution of AgNPs will also affect the response. Figure 4b shows the change in response under different particle spacings. Since the AgNPs synthesized by the Tollens method will be coated with a layer of glucose on the particle surface, this section starts the analysis from the interparticle spacing g = 2 nm. Assuming that when AgNPs are arranged in a regular and periodic array, an increase in the spacing between particles results in a blue shift and broadening of the absorption peak. The smaller the interparticle spacing, the more obvious the local effect between each other. In this study, the method can make the AgNPs densely distributed, resulting in a relatively strong local electromagnetic field enhancement.

Through simulation analysis, it is observed that changing the size of the nanoparticles cannot significantly regulate the LSPR of absorption-dominated nanoparticles; the spacing between nanoparticles is formed by self-assembly, a process in which it is difficult to achieve precise control. Therefore, this study uses the method of changing the thickness of the dielectric layer to study the optical response characteristics of AgNP-dielectric-metal structures. By changing the thickness of the dielectric layer, the cavity mode can be adjusted within a certain wavelength, thereby adjusting the driving field and effectively regulating the position of the absorption peak.

### 3.3. Effect of Synthesis Concentration

This paper first explored the effect of AgNPs synthesized with different concentrations of glucose on the response, and studied glucose solutions with concentrations of 0.1 M, 0.2 M, 0.4 M, and 1.0 M. Through the test of reflectance spectrum, AgNPs synthesized with these concentrations of glucose were observed to couple well with the response of the structure, but different glucose concentrations had a certain effect on the response. Figure 5 shows the response if the structure at concentrations of 0.1 M, 0.2 M, and 1.0 M. At a concentration of 0.4 M, the synthesized AgNPs and the underlying structure form a good coupling, and the regulation of the response is ideal. At low glucose concentrations, the protective effect of glucose on AgNPs is diminished, leading to an increased exposure of silver particles to the air, thereby affecting its performance as a metal, which is reflected in the spectrum line as the weakening of the absorption peak intensity; at the same time, when the glucose concentration is high, too much glucose will be present in the AgNP colloid, and the agglomeration of the colloid is more obvious. Since glucose is a non-conductive substance, too much glucose plays a role similar to that of a dielectric layer, which weakens the surface plasmon effect of the particles. The LSPR of the upper AgNPs cannot be well coupled with the cavity mode of the lower layer, which is reflected in the spectrum line such that the absorption peak becomes less obvious. Therefore, in this study, a 0.4 M glucose solution was used to synthesize AgNPs.

To verify that the AgNPs deposited in the functional area have good uniformity, this study randomly selected five points in the functional area and conducted spectral tests. Figure 6 shows the test results. In the functional area, the obtained signal consistency is good, indicating that the deposited AgNPs have good uniformity in the functional area.

Next, this paper tests the regulation effect of the prepared AgNP-dielectric-metal structure on the LSPR peak. In this section, the dielectric layer thickness is t_1_ = 160 nm, t_2_ = 190 nm, and t_3_ = 220 nm. The deposited AgNPs are synthesized using 0.4 M glucose. According to the previous results, the synthesized particle size is between 20 nm and 50 nm, and the diameter of the particle is set to 20 nm. When the above parameters are used, the LSPR response of the structure falls in the visible light band, and in this band, the spectrometer used for the test signal has a good characterization effect. At the same time, under this parameter, the preparation of the dielectric layer has little effect on the cavity environment of the processing equipment. Figure 7a is the normalized simulation result. As the thickness of the dielectric layer increases, the position of the absorption peak redshifts. The absorption peaks P1, P2, and P3 corresponding to t_1_, t_2_, and t_3_ are at 517.4 nm, 558.9 nm, and 604.2 nm, respectively, and have a linear relationship. Figure 7b shows the measured results: P1, P2, and P3 are 505.4 nm, 555.7 nm, and 604.4 nm, respectively. The measured results are basically consistent with the simulation. Compared with the simulation, the absorption peak of the measured results has a certain redshift and a certain broadening. This is because the deposition of AgNPs on the dielectric layer is not regular and periodic, and its distribution has a certain randomness, so the absorption peak has a certain broadening; in the previous design analysis, the increase in the size of AgNPs will lead to a redshift in the absorption peak. The actual size of the synthesized AgNPs is not strictly 20 nm but within a certain range. Therefore, in the actual measurement, the absorption peak has a slight redshift and broadening.

### 3.4. SERS Detection

Before the spectral test, this section verifies the SERS detection capability of the device when AgNPs are deposited on a specific structure. The SERS detection performance of the structure was tested by dropping a Rhodamine 6G (R6G) aqueous solution on the functional area and observing its SERS signal. The Raman shift and frequency mode assignment in the SERS spectrum of R6G are shown in Table 1.

Figure 8 shows the SERS detection signal of the AgNP-dielectric-metal structure for R6G at a concentration of 10^−4^ M at an excitation wavelength of 633 nm. It can be seen that AgNPs can be effectively deposited on the dielectric layer and self-assembled into a SERS substrate. The results of Figure 8 verify the ability of the AgNP-dielectric-metal structure as a SERS detection substrate.

## 4. Conclusions

Based on the theory and properties of surface plasmons, this paper constructs a structure of AgNP-dielectric-metal. This structure can realize SERS detection while effectively regulating the LSPR peak of the structure. This article uses the improved Tollens method to synthesize AgNPs with good surface plasmon properties and good stability. The synthesized AgNPs are deposited with the assistance of perforated tape, which effectively avoids the coffee-ring effect of liquid evaporation. In this paper, the AgNP-dielectric-metal three-layer structure constructed by AgNPs can effectively localize energy to achieve precise regulation of the absorption peak; on the other hand, AgNPs can be effectively deposited and self-assembled on the dielectric layer SiO_2_ to form a SERS substrate. By adjusting the structural parameters, the absorption peak can be effectively regulated. This paper confirms the regulation ability of the AgNP-dielectric-metal three-layer structure in actual tests. At the same time, the structure is easy to process and has good potential.

## Figures and Tables

**Figure 1 sensors-24-05778-f001:**
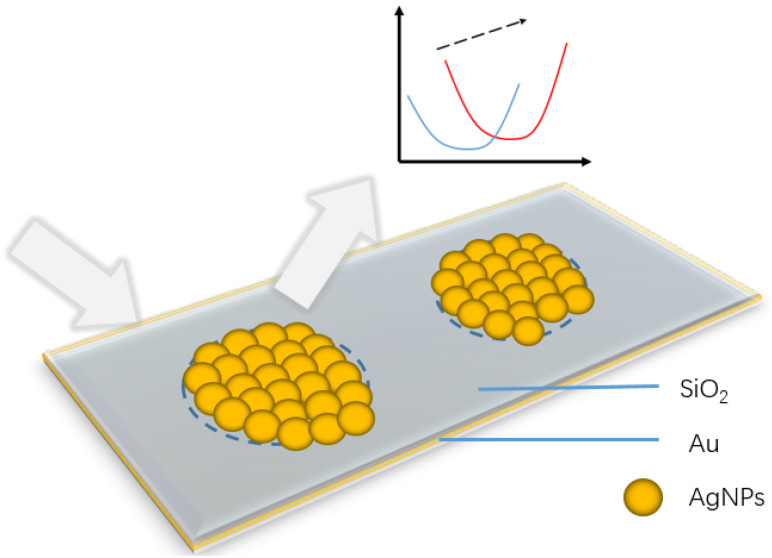
Schematic diagram of AgNP-dielectric-metal structure.

**Figure 2 sensors-24-05778-f002:**
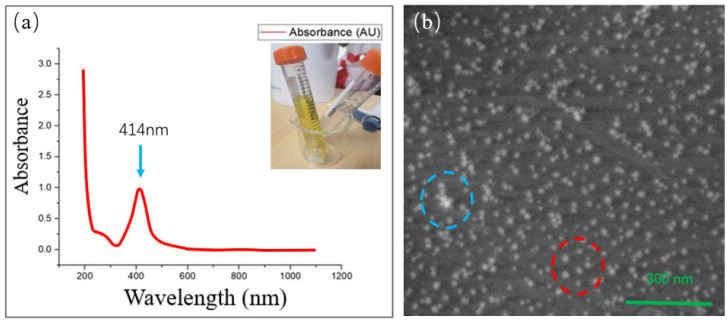
(**a**) Absorption spectrum of AgNPs synthesized using the improved Tollens method; (**b**) SEM image of the synthesized AgNPs.

**Figure 3 sensors-24-05778-f003:**
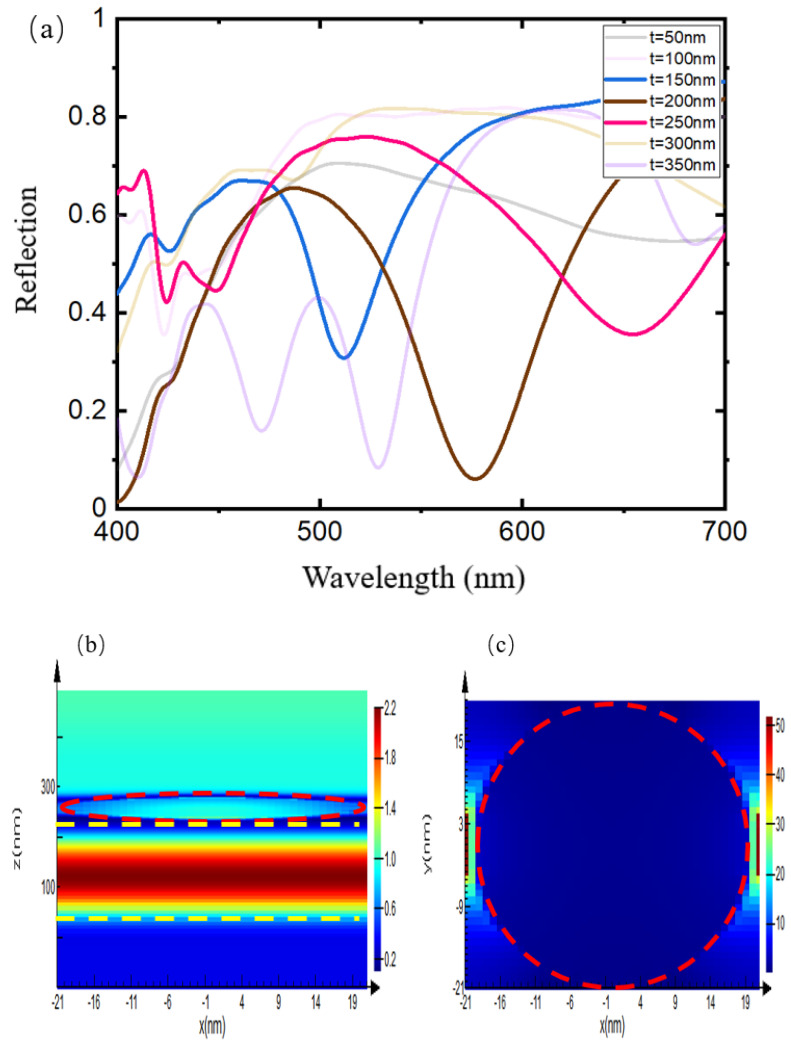
The effect of SiO_2_ layer thickness on the structural response. (**a**) SiO_2_ with different thicknesses was tested in the simulation, ranging from t = 50 nm to 350 nm. The diameter of AgNPs was set to 40 nm, and the spacing between them was 2 nm. (**b**) The electric field distribution of the structure: the energy is effectively localized in the dielectric layer (inside the yellow dashed line), the red dashed circle exhibit the AgNPs. (**c**) The electromagnetic field distribution of the top layer: the LSPR is effectively excited between the particles, and local field enhancement is generated.

**Figure 4 sensors-24-05778-f004:**
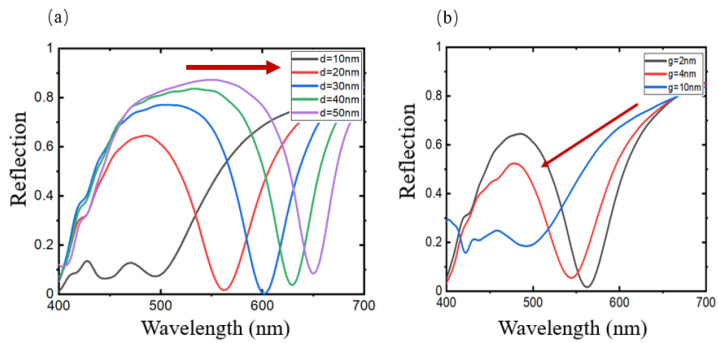
The effect of changing particle size on the response. (**a**) When the diameter of AgNPs changes from 10 nm to 50 nm, the absorption peak redshifts; (**b**) The effect of changing the particle spacing.

**Figure 5 sensors-24-05778-f005:**
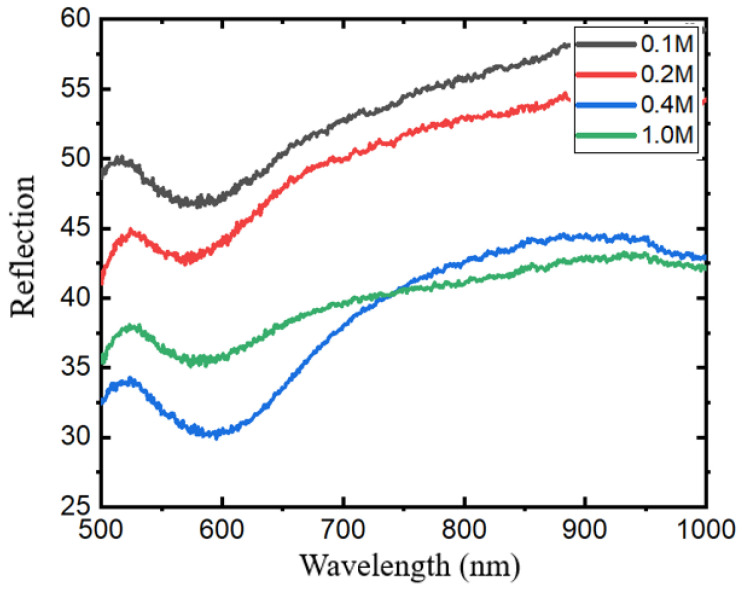
Effect of different concentrations of glucose on the structural response of AgNPs synthesized.

**Figure 6 sensors-24-05778-f006:**
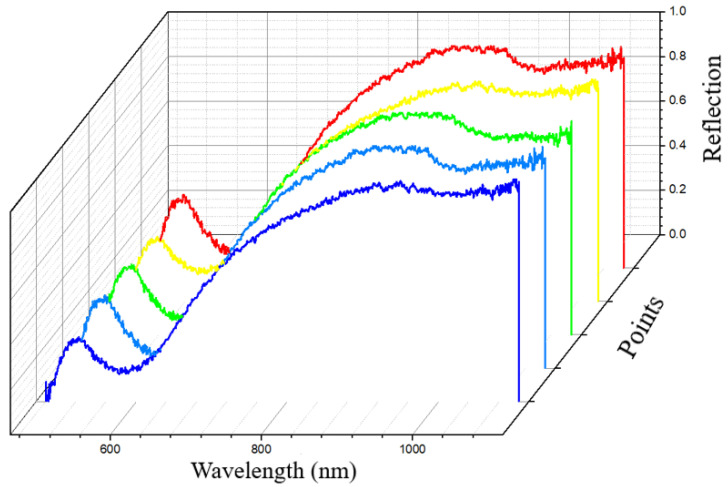
Spectral response of 5 random points in the functional area.

**Figure 7 sensors-24-05778-f007:**
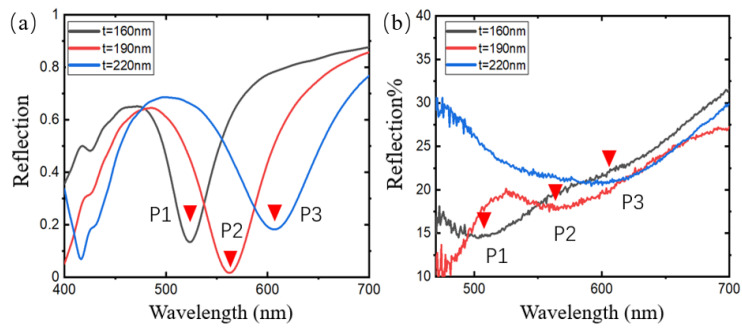
The effect of the thickness of the dielectric layer of the AgNP-dielectric-metal structure on the absorption peak. (**a**) The simulation results under three structural parameters: t_1_ = 160 nm, t_2_ = 190 nm, and t_3_ = 220 nm; (**b**) The measured results under the above three structural parameters.

**Figure 8 sensors-24-05778-f008:**
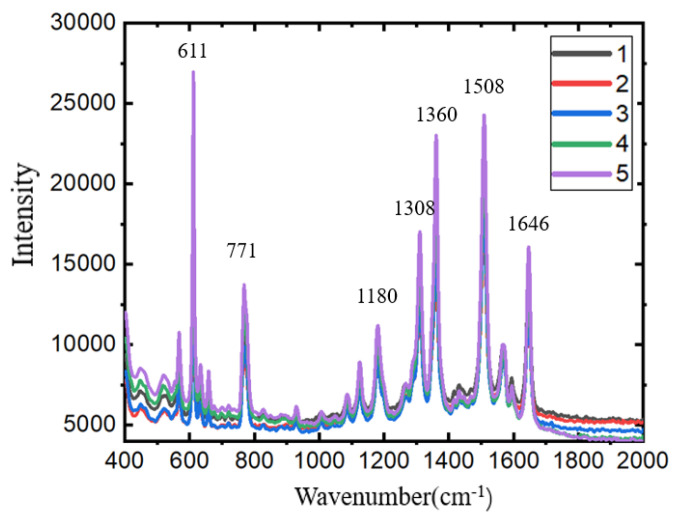
SERS detection results of AgNP-dielectric-metal structure; 1–5 are SERS signals obtained from 5 points randomly selected in the functional area.

**Table 1 sensors-24-05778-t001:** Raman shift and frequency mode assignment in the SERS spectrum of R6G [23].

Raman Shift (cm^−1^)	Assignment Mode
612	C-C-C in-plane bending vibration
774	C-H stretching
1127	C-H in-plane bending vibration
1180	C-H and N-H bending vibration
1310	C=C stretching
1364	Stretching vibration of the C-C bond
1509	Stretching vibration of the C-C bond
1574	Stretching vibration of the C=O bond
1647	Stretching vibration of the C-C bond

## Data Availability

The raw data supporting the conclusions of this article will be made available by the authors on request.

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
