# Peer review of "A Surface-Enhanced Raman Scattering Substrate with Tunable Localized Surface Plasmon Resonance Absorption Based on AgNPs"

_sensors, 2024, doi:10.3390/s24175778_

Round 1

Reviewer 1 Report

Comments and Suggestions for Authors

The manuscript titled "A SERS substrate with tunable LSPR absorption based on AgNPs" presents a novel three-layer structure for precise regulation of localized surface plasmon resonance (LSPR) absorption peaks and its application as a SERS substrate. It presents valuable insights into the design of SERS substrates and the tuning of LSPR properties. However, to enhance its quality and impact, the authors should address the following points.

1.     The LSPR property should influence the SERS enhancement effect. Detailed discussion on this aspect should be added.

2.     The long-term stability of this SERS substate should be discussed.

3.     An expanded literature review, particularly regarding the latest advancements in the stability of AgNPs and LSPR tuning mechanisms, should be included.

Reviewer 2 Report

Comments and Suggestions for Authors

This paper investigates the use of three layer structure of metal-dielectric-Ag NPs as a SERS substrate with tunable plasmonic response. The topic is of great significance because plasmonic structures for SERS applications are currently being intensively used. The authors propose to combine local plasmon resonance with Fabry–Perot modes to shift the resonance wavelength of the structure. They also demonstrate the fabrication of Ag NPs protected from the oxidization by the glucose thin layer.  

The article is written in good scientific language and all the results are presented in sufficient detail. However, there are some points that could be improved.

It is known that silver nanoparticles on a glass substrate can enhance the Raman scattering signal by an order of 108, while the process of forming silver nanoparticles on glass is easy and inexpensive. These substrates are capable to detect 10−7 M concentration of rhodamine R6G [https://doi.org/10.1186/1556-276X-7-676]. To prove the efficiency of the proposed SERS substrates, it is necessary to compare the Raman signal from R6G on a structure without Ag NPs (only gold and SiO2 layers) and estimate the enhancement factor.

It is unclear from the text if Fig.4 shows experimental or numerically simulated spectra? And are they the spectra of just Ag NPs or the whole structure?

How was the thickness of the glucose layer estimated? From the text (line 191) it appears that it was set at 2 nm, but there is no explanation.

I also have some comments to the Figures:

1.      What does the inset in Fig.1 show?

2.      Wrong figure number: Figure 2.6 appears in the text (line 164) but does not exist.

3.      In Fig.3, what does the colorbar mean? If it is a field enhancement, what is the reference level?

4.      The scale of the x-axis in Fig. 7(a) and (b) should be the same for ease of comparison of experimental and numerical results.

Reviewer 3 Report

Comments and Suggestions for Authors

The manuscript “A SERS substrate with tunable LSPR absorption based on AgNPs” examines the interaction of electromagnetic radiation in the visible region of the spectrum with the Ag nanoparticles (AgNPs) – dielectric – metal structure. This work uses the improved method to synthesize AgNPs with good surface plasmon properties. The technology of synthesis of AgNPs on a glass substrate is described in detail. SEM image of the synthesized AgNPs is shown in Fig. 2(b).  The synthesized AgNPs have a size of several tens of nanometers and are randomly distributed on a glass substrate. The article presents the results of numerical modeling of the influence of particle size and interparticle distance (Fig. 4) and dielectric film thickness (Fig. 7) on the spectra of the layered structure. The paper also presents measured spectra for different film thicknesses and concentration of glucose enveloping the nanoparticles. The structure takes advantage of the flexibility of nanoparticle application.

1)  Overall, this manuscript is interesting. However, I am not sure that I grasp the novelty of the present work. In this relation, I would suggest authors to stress novelty of their research in comparison with the previous publications. It will also be useful to compare the main obtained dependencies with the results of the studies presented in the works

- Tuning and total resonant suppression of reflection in the photonic bandgap range of Bragg reflector by two-dimensional nanoparticle array, Journal of Applied Physics,  https://doi.org/10.1063/5.0190764 ;

- Thin films by regular patterns of metal nanoparticles: tailoring the optical properties by nanodesign, Appl. Phys. B, https://doi.org/10.1007/BF01828742

2) Paper needs to be done to improve the illustrations. In Fig. 1 there is an insert with an incomprehensible graph. The main panel of Fig. 1 shows a schematic diagram of AgNPs-dielectric-metal structure, which is not identical to that discussed in the manuscript (in the Fig. 1 nanoparticles are grouped into clusters). It is not indicated for which parameters of the structure the dependences in Fig. 3(b) and Fig. 4(b) were obtained. The dependencies shown in Fig. 6 are difficult to compare, so it is recommended to present them in the same format as Fig. 5.

3) There is no description of the spectrum measurement scheme. «Figure 2.6 (b) and (c) show the simulation results of the electromagnetic field distribution of two mutually perpendicular incident lights at the absorption peak of the structure.» What purpose are “two mutually perpendicular incident lights” used for?

4) As can be seen from the SEM image (Fig. 2(b)), the synthesized AgNPs are located at some distance from each other and do not form a dense two-dimensional structure. However, in this work, simulations are performed for very dense structures (the inter-particle spacing g=2 nm). Is it correct to compare the measured values and the results of numerical modeling in this case?

5) It is necessary to add a description of the model used to obtain the dependencies in Fig. 7(a). Was the layer of glucose on the particle surface taken into account in the simulation?

6) «The absorption peaks P1, P2, and P3 corresponding to t1, t2, and t3 are at 509.7 nm, 555.7 nm, and 601.7 nm». The values given here are not accurate. From Figure 7(a) can be seen with the naked eye, the absorption peaks P1, P2, and P3 are at ~520 nm, ~560 nm, and ~605 nm.

I believe that this article can be accepted after all issues are fully resolved.

Round 2

Reviewer 3 Report

Comments and Suggestions for Authors

Some of my recommendations were not fully taken into consideration by the authors. For example, the authors did not provide any explanation for the graph shown in the inset in Fig. 1. The format of Fig. 6 does not allow comparison of the spectral response of the structure for different points, since the spectral lines are not located on the same plane.

Perhaps this manuscript can be published in its present form. However, I would like to once again draw the authors' attention to the comments that were given in point 2 of the first review. I’ll also add a request to show the boundaries of the layers of the structure in Figures 3(b),(c).
